# Virulence Bioassay of Entomopathogenic Fungi against Adults of *Atta mexicana* under Controlled Conditions

Luis J. Amaro Leal [1], Arturo Huerta de la Peña [1], Ignacio Ocampo Fletes [1], Pedro Antonio López [1], Nemesio Villa-Ruano [2,*] and Omar Romero-Arenas [3,*]

1  Colegio de Postgraduados, Campus Puebla, Boulevard Forjadores de Puebla No. 205, Santiago Momoxpan, Municipio de San Pedro Cholula, Puebla C.P. 72760, Mexico; luisjosuebiol@gmail.com (L.J.A.L.); arturohp@colpos.mx (A.H.d.l.P.); ocampoif@colpos.mx (I.O.F.); palopez@colpos.mx (P.A.L.)
2  CONAHCyT-Centro Universitario de Vinculación y Transferencia de Tecnología, Benemérita Universidad Autónoma de Puebla, Prolongación de la 24 Sur y Av. San Claudio, Ciudad Universitaria. Col. San Manuel, Puebla C.P. 72570, Mexico
3  Centro de Agroecología, Instituto de Ciencias, Benemérita Universidad Autónoma de Puebla (BUAP), Edificio VAL 1, Km 1.7 carretera a San Baltazar Tetela, San Pedro Zacachimalpa, Puebla C.P. 72960, Mexico
*  Correspondence: necho82@yahoo.com.mx (N.V.-R.); biol.ora@hotmail.com (O.R.-A.)

**Abstract:** Leafcutter ants (*Atta* spp.) are one of the most economically harmful pests in agriculture, considered dominant in the Neotropics and South America. Mature colonies of *A. mexicana* have a great economic impact on Mexico's agriculture. Microbial agents in the form of biopesticides are an effective component of integrated pest management (IPM) strategies and may present a better alternative to synthetic insecticides. Among the fungi most used as biological insecticides there are *Beauveria bassiana* and *Metarhizium anisopliae*. The objective of this research was to evaluate the effect of the entomopathogenic fungi *B. bassiana* and *M. anisopliae* of commercial origin and a native strain of *B. bassiana* from México (MA-Bb1) on adults of *Atta mexicana* under controlled conditions. In the bioassay, five formulations and a control group were tested (*B. bassiana* MA-Bb1, *B. bassiana* MA-Bb1+ Diatomin®, *B. bassiana*®, *M. anisopliae*®, Diatomin®, and Tween 80 (0.01%). The MA-Bb1+ Diatomin® biopreparation induced the highest mortality (100%) in four-week-old *A. mexicana*, followed by the MA-Bb1, *M. anisopliae*®, and *B. bassiana*® biopreparations, which caused mortality of 83.33%, 73.98%, and 68.70%, respectively. Treatments containing *B. bassiana* and *M. anisopliae* were efficient in controlling *A. mexicana* under controlled conditions. The most efficient biological control was achieved with the *B. bassiana* fungus and Diatomin®, which presented the highest total death rate in *A. mexicana* 96 h post infection, in contrast to the control group (Tween 80), which attained the lowest speed of death in the present investigation.

**Keywords:** mortality; *B. bassiana*; *M. anisopliae*; integrated pest management

## 1. Introduction

Leafcutter ants (*Atta* spp.) pose a significant threat to agriculture due to their substantial economic impact. They are recognized as dominant pests in the Neotropics [1] and South America [2,3]. In Mexico, three species of the *Atta* genus known for their leaf-cutting behavior were identified, i.e., *A. mexicana*, *A. texana*, and *A. cephalotes*. Among these, *A. mexicana* is the most widespread and numerous [4].

*A. mexicana* exhibits the intriguing behavior of cutting and collecting plant material fragments to use them as a substrate for the fungus *Leucoagaricus gongylophorus* (Möller) [5]. This symbiotic relationship enhances the efficiency of converting plant material into highly nutritious food [6]. However, these ants are also infamous as pests. Mature *A. mexicana* colonies can defoliate up to 500 kg of plant material (dry weight) annually, causing significant economic losses in agricultural, forestry, and ornamental systems across various Mexican regions [7–9].

Defoliation attributed to *A. mexicana* can occur at any stage of plant growth, demanding effective control measures to safeguard the productivity of the affected crops [10]. This presents formidable challenges, primarily due to the nocturnal defoliation behavior of adult ants, their adaptability to diverse ecosystems, and their intricate social structure [11].

Historically, the primary method of control has been the application of chemicals [12], particularly, toxic baits containing sulfluramide and fipronil as common active ingredients [13]. However, their use has declined due to environmental contamination, especially in soil and groundwater, along with issues related to their lack of specificity. Consequently, the quest for more ecologically and environmentally friendly alternatives has arisen [13]. In this context, microbial agents in the form of biopesticides have emerged as a potent component of integrated pest management (IPM) strategies, offering a potentially superior alternative to synthetic insecticides [14].

Given their promising efficacy, entomopathogenic fungi (EPF) have been extensively evaluated and recommended by numerous researchers as a biological control solution against economically significant insect pests [15,16] in diverse crop settings. These fungi possess the ability to adhere to and penetrate their hosts' cuticles, releasing enzymes such as chitinase, chitosanase, and lipase to weaken the host's regulatory system and evade its immune defenses [17]. Furthermore, they can produce compounds like beauvericin and destruxins, which paralyze insects and ultimately result in their demise [18].

Among the EPF most widely employed as biological insecticides, two stand out, i.e., *Beauveria bassiana* and *Metarhizium anisopliae*. *B. bassiana* (Ascomycota: Hypocreales) has demonstrated its ability to infect over 200 species of insects and mites spanning various genera and orders [19–22]. This fungus achieves infection by adhering to the host's cuticle through adhesive proteins, followed by the formation of an appressorium, which decomposes chitin through a combination of mechanical resistance and enzymatic degradation, facilitated by beauvericin [21].

*M. anisopliae* (Ascomycota, Hypocreales) is a generalist entomopathogen that has proven successful in controlling insect pests from more than seven orders [23]. This fungus produces various secondary fungal metabolites, including cyclic hexadepsipeptides, known as destruxins, which induce membrane depolarization by opening $Ca^{2+}$ channels, leading to the host insect paralysis and death [24].

To harness EPF effectively as biological control agents, it is imperative to develop their large-scale production and formulation systems that match or exceed the effectiveness of traditional chemicals. This is an essential prerequisite for the acceptance and commercialization of biopesticides. Additionally, the viability, growth capacity, and storage stability of these fungi must be ensured [25]. For instance, previous research tested *B. bassiana* and *M. anisopliae* against leafcutter ants [26], achieving mortality rates of 20% and 70% in *Acromyrmex* spp., and *A. sexdens* species, respectively [27]. Therefore, the aim of this study was to assess the impact of the commercially available EPF *B. bassiana*® and *M. anisopliae*®, as well as of a native *B. bassiana* strain from Mexico (MA-Bb1), on adult *A. mexicana* under controlled temperature and humidity conditions.

## 2. Materials and Methods

### 2.1. Biological Material

In this study, the native isolate of *B. bassiana* designated as MA-Bb1, with GenBank accession number MN209825.1, was used. It is maintained at the Agroecology Center of the Institute of Sciences of the Benemérita Universidad Autónoma de Puebla (ICUAP-BUAP) and was isolated from *Sphenarium purpurascens* in a corn (*Zea mays*) cultivation. The following procedure was employed. *B. bassiana* was cultivated on Sabouraud dextrose agar (SDA) at a consistent temperature of 26 ± 1 °C for seven days with a photoperiod of 12 h white light and 12 h dark. Subsequently, the culture's surface was gently scraped using a sterile glass rod, and the resulting material was mixed with 100 mL of a sterile 0.01% Tween 80 aqueous solution (Sigma, St. Louis, MO, USA). Then, the concentration of

conidia was adjusted to $1 \times 10^8$ conidia/mL by direct counting using a Neubauer chamber. The resultant suspension of 250 mL was stored at 8 °C until its application.

The commercial strains of *B. bassiana*® and *M. anisopliae*® used in this study were obtained from the collection of entomopathogenic fungi (CHE) of the National Reference Center for Biological Control (CNRCB), identified under the code keys CHE-CNRCB 497 and CHE-CNRCB 535, respectively (Liquid presentation 250 mL). The viability of the conidia was confirmed by following the protocols described by Benitez [28] and Zhang et al. [29], as follows. We placed five 10 µL aliquots of the final conidial suspension in $100 \times 15$ mm Petri dishes, with each formulation replicated in triplicate. After a 24 h incubation period, the number of germinated conidia displaying only germ tubes in the five sampling points was recorded. This value represented the percentage of germination rate for each formulation at the time of inoculation under laboratory conditions.

Additionally, diatomaceous earth was used, a material composed mainly of silicon dioxide, derived from the fossilized remains of diatom algae. Diatomaceous earth was selected due to its ability to dissolve cuticular lipids, thereby facilitating the penetration through the exoskeleton of the target insects. Furthermore, diatomaceous earth was observed to play a role in contact chemical communication processes, which could improve the efficacy of the entomopathogenic agent [30]. The commercial product was diluted to 10% (*w/v*) in distilled water (10 g Diatomin®: 100 mL water) and shaken for 5 min.

### 2.2. Virulence Bioassay

The population of *A. mexicana* utilized in this study was obtained from a colony maintained at the Benemérita Universidad Autónoma de Puebla (BUAP) Ecocampus, located at 18°56′14.0″ N 98°09′24.7″ W. These ants were approximately four months old, consisting of workers with head lengths ranging from 0.8 to 2.2 mm. They were collected in March 2022 and subsequently maintained in the mycology laboratory of the Centro de Agroecología (ICUAP-BUAP). The ants were housed under controlled conditions, including a temperature of $24 \pm 2$ °C, a relative humidity of 80%, and a photoperiod of 12 h of light, until the initiation of the bioassays.

The bioassays encompassed the evaluation of five formulations, along with a control group that featured a 0.01% aqueous solution of Tween 80. Each formulation was tested in four replicates (Table 1). Diatomin® with granules measuring less than 1 mm in size was used in combination with the MA-Bb1 strain in a single application.

**Table 1.** Biopreparation formulations.

| Treatments | Conidia/mL | g | mL |
|---|---|---|---|
| *Beauveria bassiana* MA-Bb1 | $1 \times 10^8$ | - | 250 |
| *B. bassiana* MA-Bb1+ Diatomin® | $1 \times 10^8$ | - | 250 |
| [a] *B. bassiana*® | $1 \times 10^8$ | - | 250 |
| [b] *M. anisopliae*® | $1 \times 10^8$ | - | 250 |
| [c] Diatomin® (Diatomaceous earth) | - | 10 | 100 |
| Tween 80 (0.01%) | - | - | 1000 |

[a] Code: CHE-CNRCB 497 (*Beauveria bassiana*®) https://www.gob.mx/senasica/documentos/coleccion-de-hongos-entomopatogenos (accessed on 2 February 2023); [b] Code: CHE-CNRCB 535 (*Metarhizium anisopliae*®) https://www.gob.mx/senasica/documentos/coleccion-de-hongos-entomopatogenos (accessed on 2 February 2023); [c] Code: Diatomin® https://difesa.mx/products/diatomin%C2%AE (accessed on 16 February 2023).

This combination was selected due to its recognized status of natural insecticide, widely employed in integrated pest control. Additionally, it exhibits preservative qualities for conidia, as outlined by Romero-Arenas et al. [22] and Zeni et al. [31].

The bioassays involved the application of 30 mL of each biopreparations to groups of 100 insects using a Macherey-Nagel™ chromatography sprayer (1153213-JVLAB, Guangzhou, China) following the Burgerjon methodology. [32]. The groups of insects were placed in plastic containers that were 35.9 cm long, 19.4 cm wide, and 12.4 cm high

and then incubated in a climate-controlled chamber at a constant temperature of $27 \pm 1\,°C$ and a relative humidity of $80 \pm 1\%$.

### 2.3. Preparation of the Synthetic Diet

The solid diet consisted of a mixture of bacteriological agar with 5% (*w/v*) glucose, 1% bacteriological peptone, and 0.1% yeast extract, all dissolved in 100 mL of distilled water. This mixture was heated in a microwave oven for 6 min to ensure the complete solubilization of the components, following the method of Bueno et al. [33]. The diet was then autoclaved at 120 °C for 15 min and stored in 90 mm Petri dishes in a refrigerator, following the protocol of Bueno et al. [34], until the beginning of the bioassays.

In each container with 100 insects, a plastic lid with a diameter of 1.8 cm was inserted, which contained equal portions of the solid diet with dimensions of 1 cm in length, 1 cm in width, and 0.7 cm in thickness. The diet was renewed daily for 20 days in each treatment, with the previous day's portion of diet removed when at least more than half of it had been consumed.

### 2.4. Evaluation of Insecticidal Activity

Mortality was recorded over a span of thirteen consecutive days. This time frame was selected based on prior experimental data demonstrating that ants, approximately four months old, can survive for more than twenty days when removed from colonies and provided with a solid diet [35].

The virulence assay was replicated twice, and the number of deceased ants was documented daily throughout the thirteen-day period. Subsequently, the dead insects were removed from the colony and placed in humid chambers at 20 °C, suitable to determine the cause of death and identify the insects that died due to the action of the entomopathogen (virulence). These data were used to calculate the mortality rates, employing Abbott's [36] correction method. The formula for Abbott's correction is as follows:

$$1 - [(n) \text{ in T after treatment}/(n) \text{ in Co after treatment}] \times 100 \tag{1}$$

where Co = number of control ants alive after treatment; T = number of live ants in the treated group [37].

### 2.5. Statistical Analysis

Statistical analysis of the data was conducted using analysis of variance (ANOVA), followed by a Tukey post hoc test for homogeneous groups. This analysis was performed with the statistical software Statistical Package for Social Sciences, version 17 (SPSS; IBM, Chicago, IL, USA), at a 95% confidence level.

To determine the median lethal time ($LT_{50}$) and the high lethal time ($LT_{80}$), Probit analysis was employed, following Finney's [38] methodology. This analysis expresses the probability of death (P) using the following formula:

$$P = e\hat{\ }(a + bx)/1 + e\hat{\ }(a + bx) \tag{2}$$

where P is the probability of death; *e* is the base of the natural logarithm; a and b are parameters obtained from the Probit analysis; x represents the independent variable.

This analysis allowed the calculation of the $LT_{50}$ and $LT_{80}$ values, which are important indicators of the effectiveness of a treatment in relation to the time required to achieve a certain level of mortality in ants. The Kaplan–Meier analysis was also selected to determine the survival of *A. mexicana* individuals after exposure to the biopreparations that had been applied. Finally, the specific mortality rate $K(d^{-1})$ was determined. The results were expressed in percentages which were converted using angular transformation ($\sqrt{x} + 1$). The data were fitted to an exponential decay function, following Rodríguez-Gómez et al. [39]:

$$Y = 100; \text{ IF } 0 \le t \le t0 \tag{3}$$

$$Y = (100 - S)\, e\hat{}(-k\,(t - t0)) + S; \text{ if } t > t0$$

where Y = survival percentage at time t; k is the specific mortality rate $(d^{-1})$; t0 is the death onset time (d); S is the estimated asymptotic survival level (%).

In this study, insects were collected approximately 300, 312, 384, 480, and 576 h after inoculation with each biopreparation. These insects were immersed in 4% (*v/v*) glutaraldehyde at pH 7.2 in 0.2 M phosphate buffer for 48 h at 4 °C. Subsequently, the samples were rinsed with 0.2 M phosphate buffer and subjected to a dehydration process using a gradient of ethanol solutions from 30% to 100%, with 15 min intervals between each step, using t-butyl alcohol to displace ethanol. The samples were dried using critical point drying, coated with a palladium–gold film, and examined and photographed using a scanning electron microscope JEOL JSM-6610 (Akishima, Kanto, Japan).

## 3. Results

### 3.1. Viability of Different Biopreparation Formulations

Our results revealed that the concentration of viable conidia in the MA-Bb1 strain exhibited statistically significant differences among the various biopreparation formulations ($p \leq 0.05$). The highest viability of these reproductive structures was observed in the MA-Bb1+Diatomin® formulation, with an impressive viability rate of 98.63%. In contrast, the *M. anisopliae*® formulation demonstrated the lowest percentage of viability, amounting to 68.63% (Table 2).

**Table 2.** Viability of conidia in the different biopreparations.

| Treatments | Conidia/mL * | Viability (% *) (μ ± σ) |
|---|---|---|
| *Beauveria bassiana* MA-Bb1 | $9.38 \times 10^{7\,\text{b}}$ | $93.83 \pm 0.14$ [b] |
| *B. bassiana* MA-Bb1+ Diatomin® | $9.86 \times 10^{7\,\text{a}}$ | $98.63 \pm 0.04$ [a] |
| *B. bassiana*® | $7.31 \times 10^{7\,\text{c}}$ | $73.13 \pm 0.31$ [c] |
| *M. anisopliae*® | $6.86 \times 10^{7\,\text{d}}$ | $68.63 \pm 0.02$ [d] |

* Equal letters indicate that there are no statistically significant differences ($p < 0.05$) according to Tukey's test between treatments. (μ ± σ) = mean and standard deviation.

The speeds of conidial germination and conidial viability are critical parameters associated with virulence. Virulence, in this context, was directly associated with the EPF capacity to reduce insect mortality. This finding highlights the importance of carefully selecting biopreparation formulations to improve the effectiveness of biological control agents.

### 3.2. Mortality of A. mexicana under Controlled Conditions

The results obtained in this study revealed significant statistical differences (ANOVA: F = 36.92, $p = 0.001$). Among the biopreparations assessed, the MA-Bb1+Diatomin® biopreparation exhibited the highest total mortality, achieving 100% mortality in four-week-old *A. mexicana* ants. It was followed in effectiveness by the biopreparations MA-Bb1, *M. anisopliae*®, and *B. bassiana*®, which caused mortality rates of 83.33%, 73.98%, and 68.70%, respectively (Figure 1). No mortality was observed in the control group.

In this study, Diatomin® demonstrated an insecticidal effect of 58.54% in the mortality bioassay. However, when combined with the *B. bassiana* strain MA-Bb1, the effectiveness of the entomopathogen was considerably improved.

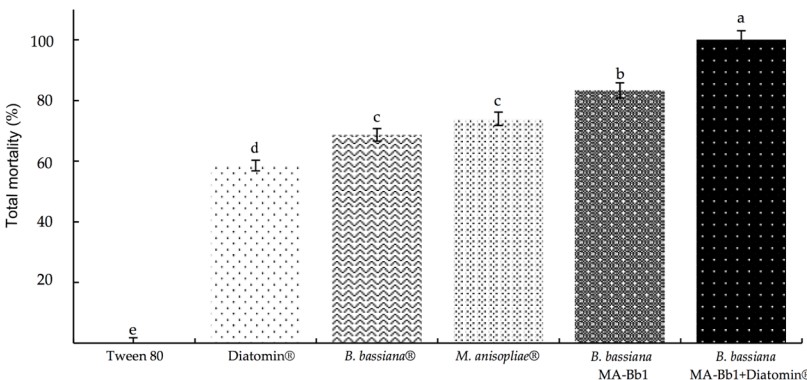

**Figure 1.** Total mortality (%) of *A. mexicana* in different biopreparations 96 h after in vitro inoculation. Equal letters indicate that there are no statistically significant differences ($p < 0.05$) according to the Tukey test between treatments. Error bars (EB) = standard deviation.

### 3.3. Lethal Time (LT)

Both EPF here studied infect their hosts until they cause the death of the organism, as mentioned above. Although infection can occur at various points on the cuticle, this study observed a preference for central and peripheral sites in *A. mexicana*.

Probit transformation was used to determine the median lethal time ($LT_{50}$) of the most virulent biopreparations. Highly significant differences were observed (ANOVA: F = 7891.20, $p = 0.001$). The estimated time required to kill 50% of the individuals was 48 h for the biopreparations MA-Bb1+Diatomin® and MA-Bb1. Conversely, the *M. anisopliae*® and *B. bassiana*® biopreparations showed a median lethal concentration ($LT_{50}$) of 72 h (Table 3), with no significant statistical differences in this case.

**Table 3.** Lethal time of *A. mexicana* when exposed to different biopreparations.

| Treatments | $LT_{50}$ (Hours *) | CI of 95% | *p* Value | $LT_{80}$ (Hours *) | CI of 95% | *p* Value |
|---|---|---|---|---|---|---|
| *Beauveria bassiana* MA-Bb1 | 48 [a] | 41.37–54.62 | 0.001 | 96 [b] | 94.7–97.2 | 0.001 |
| *B. bassiana* MA-Bb1+Diatomin® | 48 [a] | 43.34–52.65 | 0.001 | 72 [a] | 70.7–73.2 | 0.001 |
| * *B. bassiana*® | 72 [c] | 64.03–79.96 | 0.001 | 120 [c] | 118.7–121.2 | 0.001 |
| + *M. anisopliae*® | 72 [c] | 63.33–80.66 | 0.001 | 120 [c] | 118.7–121.2 | 0.001 |
| Diatomin® | 96 [d] | 89.45–102.54 | 0.001 | 120 [c] | 119.7–122.2 | 0.001 |
| Tween 80 (0.01%) | 192 [e] | 177.74–206.25 | 0.001 | 264 [d] | 262.7–265.2 | 0.001 |

* Equal letters indicate that there are no statistically significant differences ($p < 0.05$) according to Tukey's test between treatments. CI = confidence interval.

According to the results presented in Table 3, the MA-Bb1+Diatomin® biopreparation exhibited the shortest $LT_{80}$, with a duration of 72 h, followed by the MA-Bb1 treatment, which presented an $LT_{80}$ of 96 h. In contrast, the *M. anisopliae*, *B. bassiana*®, and Diatomin® treatments showed an $LT_{80}$ of 120 h each.

A strong positive correlation was observed between the variables $LT_{50}$, $LT_{80}$, and time of mycosis manifestation, indicating that treatment with the different biopreparation formulations showed a high positive correlation with the mortality of *A. mexicana* in vivo (Table 4). Additionally, survival was negatively correlated with the variables $LT_{50}$, $LT_{80}$, and time of mycosis manifestation after treatment with the different biopreparation formulations.

The shortest time of mycosis manifestation in the ants presented highly significant differences (ANOVA: F = 36.92, $p = 0.001$). The MA-Bb1+Diatomin® biopreparation achieved mycelial coating 312 h post inoculation under in vitro conditions, followed by the *B. bassiana* (MA-Bb1) biopreparation, which induced mycelial coating 384 h after inoculation. The *M. anisopliae*® biopreparation required the shortest time (576 h) to induce mycelial coating in *A. mexicana*.

**Table 4.** Correlation of the different variables evaluated.

| | Variables | | (1) | (2) | (3) | (4) | (5) | (6) |
|---|---|---|---|---|---|---|---|---|
| (1) | $LT_{50}$ | Pearson correlation | 1 | 0.950 ** | 0.908 ** | −0.908 ** | 0.805 ** | 0.825 ** |
| | | Covariance | | 2898 | 1197 | −1197 | 6801 | 69 |
| (2) | $LT_{80}$ | Pearson correlation | | 1 | 0.953 ** | −0.953 ** | 0.828 ** | 0.782 ** |
| | | Covariance | | | 1650 | −1650 | 9201 | 87 |
| (3) | Mortality | Pearson correlation | | | 1 | −1.000 ** | 0.865 ** | 0.807 ** |
| | | Covariance | | | | −747 | 4149 | 38 |
| (4) | Survival | Pearson correlation | | | | 1 | −0.865 ** | −0.807 ** |
| | | Covariance | | | | | −4149 | −38 |
| (5) | Time of mycosis manifestation | Pearson correlation | | | | | 1 | 0.956 ** |
| | | Covariance | | | | | | 294 |
| (6) | Treatment | Pearson correlation | | | | | | 1 |

** The correlation is significant at the 0.01 level.

Studies carried out with a scanning electron microscope (SEM) provided information on the infection and colonization of *A. mexicana* by EPF. The observations included the identification of a mycelial network with conidial balls (Figure 2A,B), featuring globose conidia arranged on the mesosome's cuticle. This characteristic is typical of *B. bassiana* and was observed across the insect's body, resulting in a mummified appearance of the host.

The microscopic analysis also confirmed the adherence of the Diatomin® treatment to most of the insect's body, with a more pronounced presence around the head of *A. mexicana* (Figure 2C) and the compound eye (Figure 2D). Additionally, extensive networks of *B. bassiana* hyphae and a high mycelium density were observed in the Diatomin® treatment on the body surface of *A. mexicana*. These observations indicated signs of fungal vegetative growth after 312 h of treatment (Figure 2E,F). *B. bassiana* formulations involved the processing of effective agents and other components in specific proportions.

On another note, SEM analysis after 576 h of treatment revealed the presence of the entomopathogenic fungus *M. anisopliae*® across most of *A. mexicana* femur cuticle. It was organized into a robust mycelial network covering the entire surface (Figure 3), with green spores observed on the insect's cuticle.

The MA-Bb1+Diatomin® biopreparation, which showed high viability of the conidia, caused higher mortality in four-week-old *A. mexicana* compared to the other preparations. It is important to highlight that, considering that all formulations started from the same initial concentration of conidia, these results suggest that Diatomin® can preserve the concentration and viability of *B. bassiana* (MA-Bb1) conidia at a temperature of 26 °C ± 1 and a relative humidity of 84% ± 2.

The results obtained in this study revealed significant statistical differences in the cumulative survival of four-week-old *A. mexicana* exposed to the different biopreparations. Among the biopreparations evaluated, the biopreparation MA-Bb1+Diatomin® exhibited the lowest cumulative survival 96 h after inoculation. This was followed by the biopreparations MA-Bb1, *M. anisopliae*®, and *B. bassiana*® and Diatomin®, associated with cumulative survival values of 123, 144, and 170 h, respectively (Figure 4). The cumulative survival of the control group was greater than 300 h after inoculation.

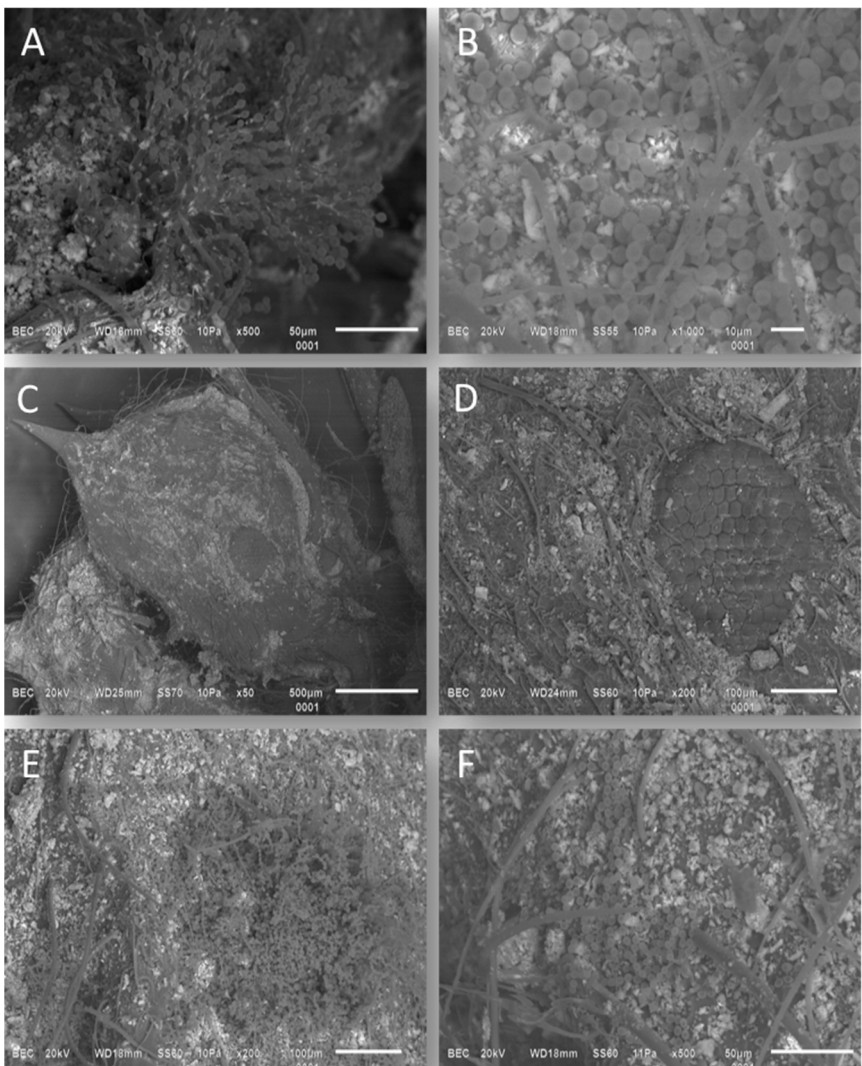

**Figure 2.** Representative scanning electron micrographs (SEM) of the infection by EPF and Diatomin®️ in *A. mexicana*. (**A**,**B**) Conidia *B. bassiana* (MA-Bb1) fungi displayed on the mesosome cuticle. Effect of Diatomin®️ on the head of *A. mexicana* (**C**) and around the compound eye (**D**). Effect of combined *B. bassiana* (MA-Bb1) and Diatomin®️ at the end of the host's abdomen and (**E**) in the intersegmental folds on the abdomen (**F**).

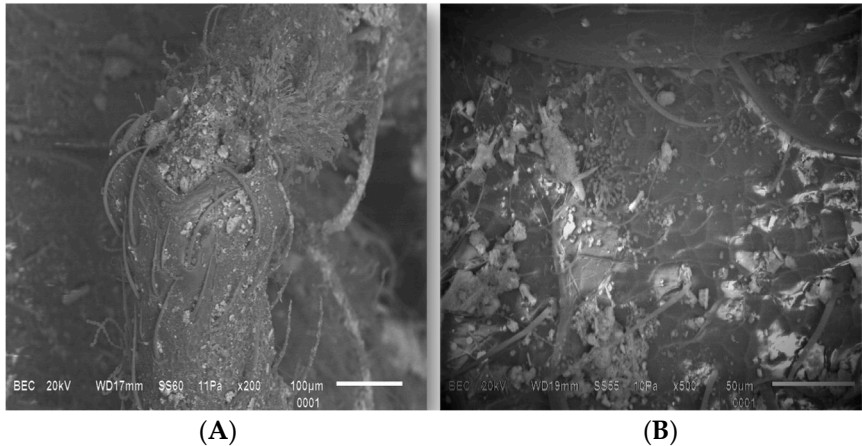

**Figure 3.** Scanning electron micrographs (SEM) representative of the infection process by commercial fungi. (**A**,**B**) Fungal conidia of *M. anisoplaie* ®️ visualized on the femur of *A. mexicana*.

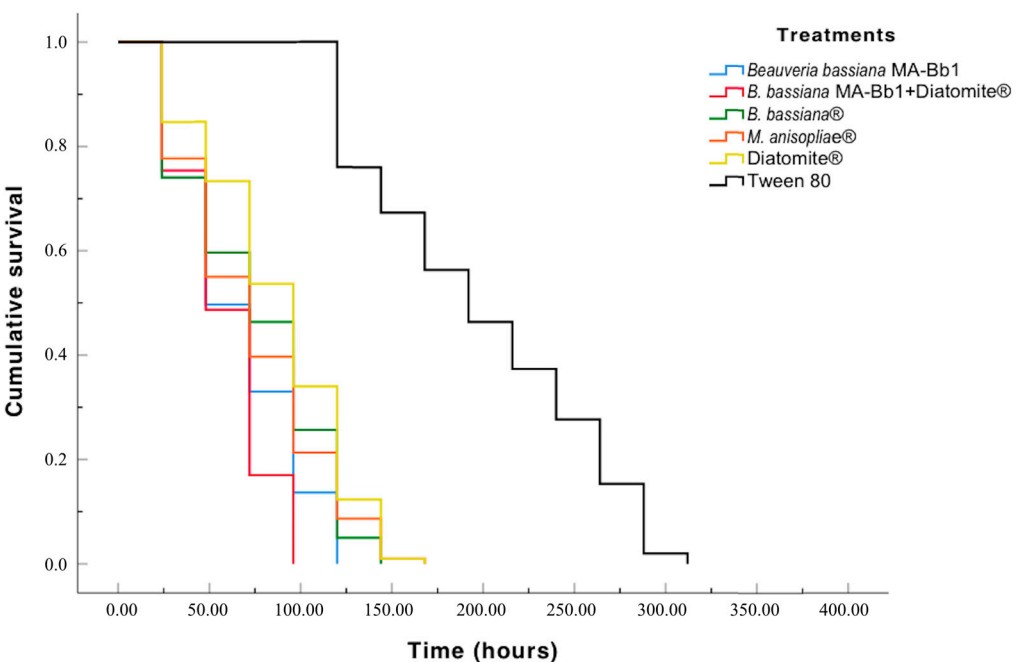

**Figure 4.** The cumulative survival analysis revealed statistically very significant effects of the different biopreparations (log rank: $\chi^2$ = 941.628, *p* = 0.0001) in *A. mexicana* during 15 days of treatment.

The variation in $LT_{50}$ and cumulative survival of four-week-old *A. mexicana* exposed to the different biopreparations may be associated with the infection and penetration capacity of the fungi, the susceptibility of the host, and the viability of germination of the conidia.

## 4. Discussion

In this study, we evaluated the feasibility of different biopreparation formulations and their effectiveness in controlling the mortality of *A. mexicana* ants.

The entomopathogenic fungus *B. bassiana* plays a crucial role as a biological control agent against a wide range of agricultural and forestry pests [40]. It has gained approval for commercial use in various countries [41,42].

The speed of conidial germination and conidial viability are critical parameters associated with its virulence. Virulence, in this context, is directly associated with EPF capacity to reduce insect mortality, even in the presence of host resistance [43]. Notably, conidial germination is an indispensable process for initiating infection by *B. bassiana* [44].

The genus *Metarhizium* has been well documented as an effective biocontrol agent against a wide range of insect pests, including leafcutter ants [45–47]. Similarly, the *Beauveria* genus is known for producing secondary metabolites with insecticidal properties, such as beauvericin [48]. Furthermore, the genetic variability present in EPF may be one of the factors responsible for the differences in virulence between *B. bassiana* and *M. anisopliae* [49].

Among the biopreparations evaluated, the MA-Bb1+Diatomin® biopreparation exhibited the highest total mortality, achieving 100% mortality in four-week-old *A. mexicana* ants. These values are higher compared to those reported by Ribeiro et al. [50], who obtained a mortality of 50% in *A. sexdens* on the third day after inoculation with a suspension of *B. bassiana* at a concentration of $1.0 \times 10^9$ conidia/mL. Choe et al. [51] mentioned that Diatomin® plays a role in dissolving cuticular lipids, facilitating the penetration of fungi through the exoskeleton, and is involved in chemical contact communication processes [52]. Song et al. [53] conducted a study showing that the combination of surfactants like zeolite, perlite, and vermiculite with fungal conidia of *B. bassiana* and *M. anisopliae* exhibited high virulence against *Riptortus pedestris* in vivo. Moreover, Akbar et al. [54] reported that diatomaceous earth increased the mortality of *T. castaneum* larvae. In addition, Vassilakos et al. [55] reported an additive effect of *B. bassiana* when used in conjunction

with diatomaceous earth against adult *R. dominica* and *S. oryzae*, findings consistent with those reported in this investigation.

Furthermore, our Probit analysis revealed significant differences in the median lethal time (LT$_{50}$) between the biopreparations. These findings underscore the importance of considering the rate of action of biological control agents when evaluating their effectiveness in the field. Probit analysis indices such as mean lethal time (LT$_{50}$) are commonly used parameters to evaluate the effectiveness of pest control agents [56]. These findings are similar to those of Quesada-Moraga et al. [57], who observed 100% mortality nine days (216 h) after the application of *M. anisopliae* against *C. capitata*. This positive association is a phenomenon that was explained by several researchers [58–60] in the majority of entomopathogenic microbial fungi tested. In this sense, the entomopathogenic capacity is used in the control of social insects such as leafcutter ants [45].

A suitable mean lethal concentration increases the likelihood that an individual infected with either *B. bassiana* or *M. anisopliae* will cause the development of mycosis within the colony, thereby reducing the colony's ability to respond to the infection [61,62]. *M. anisopliae* is a facultative entomopathogen with a broad host range [63]. The conidia of this fungus infect insects by forming appressoria and producing extracellular proteolytic enzymes such as subtilisin (PR1), metalloproteases, and enzymes with chitinolytic activity [60,64,65]. This phenomenon aligns with observations made by Ortiz and Keyhani [66] and may be common in several insect species. During this infection process, fungi produce specialized infection structures, such as penetration pegs and appressoria, which allow growing hyphae to break through the host's integument. It is worth noting that the insect cuticle is a very heterogeneous structure, and its composition can vary significantly, even during different life stages of a particular insect [67].

Our findings align with the research of Vestergaard et al. [68] and Wang et al. [69], who observed that most EPF produce mycelial networks on the insect cuticle after inoculation (Figure 2). In this sense, dead corpses become the source of secondary infection and increase the effectiveness of EPF in the control of insect pests [70]. That is, at the time of producing asexual spores, known as conidia, they serve as infectious propagules contributing to the fungus's spread [71].

Our results also support the idea that the combination of EPF with agents such as Diatomin® can improve the effectiveness of entomopathogenic fungi, as observed in other studies highlighting the synergistic effects of these agents and EPF in specific formulations in pest control. Wang et al. [72] developed a wettable *B. bassiana* powder that exhibited a high control on *Frankliniella occidentalis*, achieving a control rate exceeding 74%. Xiao et al. [67] effectively used a formulation of *B. bassiana* conidia in combination with imidacloprid to control the false-eyed leafhopper [73].

The infection of *M. anisopliae* in host insects is, in general, a complex process that involves several stages, such as the adhesion of conidia through hydrophobic interactions, their germination and formation of germ tubes, the production of appressoria, the penetration in the host, colonization, hyphal extrusion, and conidiogenesis [74]. Importantly, *M. anisopliae* produces only one germ tube from each conidium [75]. Similarly, in the present study, it was observed that the conidia of *M. anisopliae*® germinated and produced a single germ tube after 48 h in four-month-old adults of *A. mexicana* under controlled conditions.

Finally, the survival results of this study indicated greater treatment effectiveness compared to those of Dornelas et al. [76], who reported that after ten days of evaluation, all *A. sexdens* workers infected with *M. anisopliae* showed a mortality rate of only 25% when a suspension of $1.0 \times 10^7$ conidia/mL was applied.

## 5. Conclusions

The findings in laboratory conditions of this study are indeed promising, suggesting that formulations containing *B. bassiana*® and *M. anisopliae*® can be effective in controlling *A. mexicana* within a controlled environment.

The most efficient biological control observed in this study was achieved with the combination of *B. bassiana* (MA-Bb1) and Diatomin®, which demonstrated the highest total mortality rate of *A. mexicana* 96 h post infection, in contrast to the control group (Tween 80), which exhibited the lowest specific mortality rate in the study.

It is important to note that results obtained in laboratory settings often serve as a starting point for assessing the efficacy of treatments. However, field trials are essential to determine their practical applicability in real-world scenarios. Additionally, the effectiveness of these treatments may vary depending on the insect species and environmental conditions. Therefore, further studies are necessary to validate and fine-tune these biological control strategies before considering their large-scale implementation.

**Author Contributions:** Conceptualization, L.J.A.L., I.O.F. and O.R.-A.; methodology, L.J.A.L., N.V.-R. and O.R.-A.; software, N.V.-R. and O.R.-A.; validation, I.O.F., A.H.d.l.P. and O.R.-A.; formal analysis, L.J.A.L., P.A.L. and O.R.-A.; resources, L.J.A.L. and O.R.-A.; original—draft preparation, N.V.-R., I.O.F. and O.R.-A.; writing—review and editing, I.O.F., O.R.-A. and P.A.L.; visualization, O.R.-A. and N.V.-R.; supervision, A.H.d.l.P., project administration, O.R.-A.; funding acquisition, O.R.-A. All authors have read and agreed to the published version of the manuscript.

**Funding:** The authors are grateful to Consejo Nacional de Humanidades, Ciencia y Tecnología (Conahcyt) (No. CVU 775050), Colegio de Postgraduados, Campus Puebla (No. ID 21931001) and Benémerita Universidad Autónoma of Puebla (No. ID 100420500).

**Institutional Review Board Statement:** Not applicable.

**Informed Consent Statement:** Not applicable.

**Data Availability Statement:** The raw data supporting the conclusions of this article will be made available by the authors on request.

**Conflicts of Interest:** The authors declare no conflicts of interest.

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
