# Peer review of "Virulence Bioassay of Entomopathogenic Fungi against Adults of Atta mexicana under Controlled Conditions"

_applsci, doi:10.3390/app14073039_

Round 1

Reviewer 1 Report

Comments and Suggestions for Authors

The manuscript entitled Virulence Bioassay of Entomopathogens’ Fungi Against Adults 2 of Atta mexicana Under Controlled Conditionsillustrates the effect of the entomopathogenic fungi Beauveria bassiana and Metarhizium anisopliae of commercial origin and a native strain of B. bassiana from México (MA-Bb1) against adults of Atta mexicana under controlled conditions. 

Overall, the manuscript is well written with innovations and has all merits to be published in the submitted journal.  

However, there are minor issues need to be taken care of before it can be considered for publication. My specific comments are appended below- 

In the title: “entomopathogens’ fungi” should be replaced with “entomopathogenic fungi”

Table 3: replace “p valor” with “p value”

Table 3: write “CI of 95%” instead of “IC of 95%”

Line no 266: Atta Mexicana should be in the italic form

Author Response

Thank you for your comments.

All changes were made as requested. You can see the changes in yellow in the document.

Reviewer 2 Report

Comments and Suggestions for Authors

Title: In my opinion, the word "Entomopathogens" should be changed to "Entomopathogenic" in the title.

 Introduction: Since the term "entomopathogenic fungi" is often used in the text, I propose to use the abbreviation EPF.

Line 77 - the authors use old and outdated taxonomic terms!!!, currently the correct name is M. anisopliae (Ascomycota, Hypocreales).

Materials and Methods: line 99 - please specify what the B. bassiana strain was isolated from. line 101 - in what lighting conditions was the cultivation carried out? line 133 - is the term "different formulations" appropriate here? The same applies to the title of table 1.

Line 158 - the description of "Evaluation of Insecticidal activity" lacks information whether dead insects were removed from the colony and placed in wet chambers with appropriate humidity to determine that the cause of death was a fungal infection? If not, it created a risk of secondary infections, which blurs the picture of pathogenicity.

Results and discussion: The title of Figure 1 lacks information whether it is "Total mortality" and after how many days it was diagnosed? Line 263 - the term "Lethal time" is usually abbreviated as "LT" and not "TL"!!! . under table 3 there is no information what the terms IC and "P Valor" mean. Line 371 - 373 - a trivial phrase that is obvious and previously mentioned in the introduction. I am very surprised by the fact that the authors completely confused the captions of the photos in Figure 3. I have no doubt that the photo of Ai B is the structure of M. anisopliae and B and C could be B. bassiana, but is it really? Maybe it's Aspergillus?

References: the literature list is not in order. For example, the cited item (line 267) by Ortiz and Keyhani [54] - in the list of references it is not number 54, there is Vassilakos et al. and there are more such examples, e.g. Xiao et al. [55] and so on.

Author Response

Thank you for your comments.

All changes were made as requested. You can see the changes in blue in the document.

Questions:

I am very surprised by the fact that the authors completely confused the captions of the photos in Figure 3. I have no doubt that the photo of Ai B is the structure of M. anisopliae and B and C could be B. bassiana, but it is is it really? Maybe it's Aspergillus?

Answer:

The marked error is corrected and the figure is removed to avoid misinterpretation of the images.

Questions:

References: The literature list is not in order.

Answer:

The marked error is corrected.

Reviewer 3 Report

Comments and Suggestions for Authors

Dear author, attached my suggestions (inside the document) for their manuscript, 

Author Response

Thank you for your comments.

All changes were made as requested. You can see the changes in purple color in the document.

Questions:

again LT50 and LT80????? why???? witch is the difference...

Answer:

If there is a difference between these variables.

LT50 (50% mean lethal time) and LT80 (80% mean lethal time) are terms commonly used in toxicology and pest control studies to describe the time required for a certain percentage of individuals in a population to die as a result. from exposure to a toxic agent or treatment.

The main difference between LT50 and LT80 lies in the mortality percentage to which they refer.

Reviewer 4 Report

Comments and Suggestions for Authors

The article Virulence Bioassay of Entomopathogens’ Fungi Against Adults of Atta mexicana Under Controlled Conditions assessed the effect of the entomopathogenic fungi B. bassiana and M. anisopliae of commercial origin and a native strain of B. bassiana from México (MA-Bb1) against adults of Atta mexicana under controlled conditions. I have some comments to improve it.

Introduction:

1) Lines 46-47: Same text in the abstract (Line 18).

2) The hypothesis of the study is missing.

Material and Methods:

- It is necessary to double-check if the model's assumptions have been attempted. In general, insect mortality data does not follow a normal distribution. I would recommend the use of a GLM with binomial distribution.

Line 202: use “and” instead of “y”

Results and discussion:

This section needs to be better organized. Please make a separation between results and discussion.

Conclusion

Lines 406-409: This sentence is unnecessary; it looks like a discussion.

Author Response

Thank you for your comments.

All changes were made as requested. You can see the changes in green in the document.

Questions:

The hypothesis of the study is missing.

Answer:

The hypothesis is implicit in the objectives of the work

Questions:

It is necessary to double-check if the model's assumptions have been attempted. In general, insect mortality data does not follow a normal distribution. I would recommend the use of a GLM with binomial distribution.

Answer:

For this study, the methodology proposed by Abbott, W.S. A method of computing the effectiveness of an insecticide. J Am Mosquito Contr. 1987, 3, 301–302. PMID: 3333059. We consider this methodology to be very complete and suitable for this study, and it is also the most reported in similar works.

Questions:

Results and discussion:

This section needs to be better organized. Please make a separation between results and discussion.

Answer:

We believe that the way the work is presented is the best for readers; furthermore, the journal does not require separate results and discussion and can be presented in this format.

Some examples from the magazine:

https://doi.org/10.3390/app14010466
https://doi.org/10.3390/app14010418
https://doi.org/10.3390/app132413003
https://doi.org/10.3390/app13105941

Round 2

Reviewer 2 Report

Comments and Suggestions for Authors

In my opinion, the authors have taken into account all my previous comments and the manuscript can be published in the present version.

Author Response

Thanks for the observations

Reviewer 3 Report

Comments and Suggestions for Authors

Dear authors, your manuscript has improved substantially, only a little adjustment is needed (difference in cumulative survival) references scientific names. see inside of the document.

Author Response

(The authors gave the same response as above.)

Reviewer 4 Report

Comments and Suggestions for Authors

The authors did not show progress in the paper. Therefore I recommend reject it.

Author Response

Thanks for the observations

According to their comments, the separation of the discussion and the results was carried out.

Additionally, the study was complemented with the Kaplan-Meier analysis to determine the survival of A. mexicana individuals after exposure to the applied biopreparations.

We believe that these modifications address the suggested comments.

Kind regards

Round 3

Reviewer 4 Report

Comments and Suggestions for Authors

The article can be published now.

Author Response

Thanks for the observations